# PGC-1α4 Interacts with REST to Upregulate Neuronal Genes and Augment Energy Consumption in Developing Cardiomyocytes

**DOI:** 10.3390/cells11192944

**Published:** 2022-09-20

**Authors:** Tomi Tuomainen, Nikolay Naumenko, Maija Mutikainen, Anastasia Shakirzyanova, Sarah Sczelecki, Jennifer L. Estall, Jorge L. Ruas, Pasi Tavi

**Affiliations:** 1A.I. Virtanen Institute for Molecular Sciences, University of Eastern Finland, 70210 Kuopio, Finland; 2Institut de Recherches Cliniques de Montréal, Montréal, QC H2W 1R7, Canada; 3Molecular and Cellular Exercise Physiology, Department of Physiology and Pharmacology, Biomedicum, Karolinska Institute, 17165 Stockholm, Sweden

**Keywords:** energy metabolism, transcription, heart failure, Na/K-ATPase, electrophysiology, calcium

## Abstract

Transcriptional coactivator PGC-1α is a main regulator of cardiac energy metabolism. In addition to canonical PGC-1α1, other PGC-1α isoforms have been found to exert specific biological functions in a variety of tissues. We investigated the expression patterns and the biological effects of the non-canonical isoforms in the heart. We used RNA sequencing data to identify the expression patterns of PGC-1α isoforms in the heart. To evaluate the biological effects of the alternative isoform expression, we generated a transgenic mouse with cardiac-specific overexpression of PGC-1α4 and analysed the cardiac phenotype with a wide spectrum of physiological and biophysical tools. Our results show that non-canonical isoforms are expressed in the heart, and that the main variant PGC-1α4 is induced by β-adrenergic signalling in adult cardiomyocytes. Cardiomyocyte specific PGC-1α4 overexpression in mice relieves the RE1-Silencing Transcription factor (REST)-mediated suppression of neuronal genes during foetal heart development. The resulting de-repression of REST target genes induces a cardiac phenotype with increased cellular energy consumption, resulting in postnatal dilated cardiomyopathy. These results propose a new concept for actions of the PGC-1α protein family where activation of the *Pgc-1α* gene, through its isoforms, induces a phenotype with concurrent supply and demand for cellular energy. These data highlight the biological roles of the different PGC-1α isoforms, which should be considered when future therapies are developed.

## 1. Introduction

In cardiac myocytes, as in other metabolically active cells, peroxisome proliferator-activated receptor (PPAR) gamma coactivator 1-alpha (PGC-1α) is an essential factor promoting gene expression pathways that support cellular energy production [1]. Downregulation of cardiac PGC-1α is associated with heart pathologies such as load-induced hypertrophy [2] and PGC-1α knockout in the mouse heart leads to cardiac energy deficiency and impaired heart function [3,4]. On the other hand, robust PGC-1α overexpression in the mouse heart induces overt mitochondrial proliferation [5], whereas moderate overexpression induces exercise-like changes in excitation-contraction coupling [6]. PGC-1α mediates its effects on metabolism via interaction with transcription factors such as PPARα and nuclear respiratory factor 1 [7,8]. Additionally, PGC-1α can directly regulate genes supporting cardiomyocyte contractile function by coactivating MEF2 transcription factors [9].

In addition to the originally described PGC-1α protein, termed PGC-1α isoform 1 (PGC-1α1), several other isoforms arising from differential promoter usage and splicing of the *PGC-1α* gene have been reported [10]. Interestingly, the non-canonical isoforms regulate transcriptional programs distinct from canonical PGC-1α1 in a cell type-specific manner. In skeletal muscle, PGC-1α4, a truncated isoform originating from the alternative promoter, is induced in exercise and regulates the expression program leading to skeletal muscle hypertrophy [11]. In the liver, PGC-1α4 has been shown to have anti-apoptotic effects in inflammatory situations [12]. In skeletal muscle, transcripts produced from the alternative promoter are robustly induced by β-adrenergic stimulation [13], which suggests a role for them in the heart where β-adrenergic signaling is a central regulator of cardiomyocyte function.

Despite the extensive research on PGC-1α and heart function, so far, the focus has been on the canonical PGC-1α1 isoform. One report describes downregulation of truncated N-terminal PGC-1α in myocardial infarction-induced heart failure in mice [14], but expression patterns and biological roles of the non-canonical isoforms have been overlooked in cardiac physiology. An emerging role for other isoforms in other metabolically active cell types warrants a detailed investigation of their expression and function in cardiac myocytes. In this study, we assessed cardiac *PGC-1α* expression and detected *Pgc-1α4* as the main non-canonical isoform expressed in the heart. Detailed cardiac phenotyping of a mouse model with cardiomyocyte-specific PGC-1α4 overexpression suggests RE1-Silencing Transcription factor (REST) as a novel molecular target for the PGC-1α4 protein. In developing cardiomyocytes, this interaction induces neuronal gene expression, leading to distinct changes in cell electrophysiology and increased energy consumption. In vivo, the cardiomyocyte-specific PGC-1α4 transgenics develop postnatal dilated cardiomyopathy and early death.

## 2. Materials and Methods

### 2.1. Gene Expression Analysis from RNA Sequencing Data

RNA sequencing data generated in this study or downloaded from NCBI’s GEO database (see BioProject identifiers in Appendix A) were processed as follows. Sequencing reads were aligned to genomic DNA with Hisat2 software (version 2.1.0) [15] using indexes for mouse GRCm38 or human GRCh38 genome builds. Alignments stored in bam file format were counted with QoRTs software(version 1.3.0) [16] against reference gene annotations. Ensembl’s gene annotation release version 92 was used for total gene read counting and differential gene expression was analysed with Deseq2 software (version 1.22.0) [17]. Expression data from the REST knockout model were analysed using Deseq2 software for the RNA sequencing count data available from the NCBI entry (GEO accession GSE80378). Gene annotations were compiled with Stringtie software (version 1.3.4d) [18] using alignments from datasets DS4 (mouse) and DS5 (human) in order to acquire Pgc-1α junction counts from public datasets. Gene enrichment analysis for differentially expressed genes was performed with DAVID software (version 6.7) [19].

### 2.2. Ethical Statement

All animal experiments were carried out with authorization from The National Animal Experiment Board of Finland (animal experimentation permit ESAVI/7867/2018) and following the guidelines of The Finnish Act on Animal Experimentation, which comply with the guidelines from Directive 2010/63/EU of the European Parliament on the protection of animals.

### 2.3. Experimental Animals

Non-transgenic cardiomyocytes were isolated from standard C57BL/6 strain mice. ^LSL^PGC-1α4 mice [12] were crossed with mice carrying the *cre* transgene under the Myh6 gene promoter [20] to generate mice with cardiomyocyte-specific overexpression of PGC-1α4 (α4^+^). PGC-1α knockout mice were generated by crossing Myh6-cre mice with a strain carrying floxed *Pgc-1α* as previously [4]. All experiments with transgenic animals were performed from a single mixed-background colony with age-matched littermates as controls. α4^+^ mice suffered from a severe cardiac phenotype, and they were euthanized right after the appearance of symptoms of heart failure (shortness of breath or inactivity) as per the guidelines of the experimental animal permit. The day of euthanasia was considered as the day of death in the survival analysis. Animals were kept in standard housing conditions in The National Laboratory Animal Centre of the University of Eastern Finland. Animals were fed ad libitum. All animal experiments were conducted during the daytime (from 8 a.m. to 3 p.m.).

### 2.4. Cardiomyocyte Isolation and Culture

Adult mouse ventricular myocytes at the age of four or 18 weeks were obtained by enzymatic dissociation (AfCS Procedure Protocol PP00000125) and neonatal ventricular cardiomyocytes 1–2 days after birth as in previous research [6]. See detailed methods in Appendix A online.

### 2.5. Real-Time Quantitative PCR

For RNA isolation, frozen tissue samples were homogenized with TissueLyzer II (Qiagen) into TRI reagent (Sigma-Aldrich, St Louis, MO, USA). From cultured cells, RNA was isolated using E.Z.N.A.^®^ Total RNA Kit (Omega Bio-Tek Inc., Norcross, GA, USA). cDNA synthesis was carried out with a RevertAid RT Reverse Transcription Kit (ThermoFisher Scientific, Waltham, MA, USA) and a StepOnePlus™ Real-Time PCR System (Applied Biosystems, Foster City, CA, USA) was used for quantitative PCR analysis using either TaqMan™-based chemistry with Maxima Probe/ROX qPCR Master Mix (ThermoFisher Scientific, Waltham, MA, USA) or SYBR™ Green-based chemistry with FastStart Universal SYBR^®^ Green Master (Roche, Mannheim, Germany). A standard curve for mRNA quantification from each set of samples was done using a dilution series of pooled samples. The geometrical mean of mRNA quantities determined for ribosomal RNA 18s and Beta-2 microglobulin was used to normalize the expression values of genes of interest. For the data presentation, all real-time quantitative PCR (RT-qPCR) results were normalized to the mean of the experimental control group. Sequences of custom designed primers and fluorogenic probes are listed on Appendix A.

### 2.6. Agarose Gel Electrophoresis PCR

To differentiate between different *Pgc-1α* exon 7b-containing transcripts, PCR producing long (>800 bp) products must be performed, which is not applicable to RT-qPCR. To analyse the presence of these transcripts, cDNA synthesized from the RNA that was isolated from isoprenaline-exposed adult mouse cardiomyocytes was used as template for PCR performed with Phusion™ High-Fidelity DNA Polymerase (Thermo Scientific, Waltham, MA, USA) and PCR products were analysed using agarose gel electrophoresis. Primers used in the reactions are listed on Appendix A.

### 2.7. Western Blot

The nuclear protein fraction or total cellular protein from 4-week-old mouse ventricular tissue was isolated as previously from the adult mouse heart [6]. PGC-1α protein expression was assessed from the nuclear fraction. AMPK and NKA isoform expression were assessed from the total cellular protein. Mouse anti-PGC-1α (product number ST1202, Millipore, Temecula, CA, USA), mouse anti-NKAα1 (A275, Sigma-Aldrich, St Louis, MO, USA) and mouse anti-NKAα3 (MA3-915, Thermo Fisher Scientific, Waltham, MA, USA) antibodies were used together with secondary HRP conjugated goat anti-mouse IgG antibody (R&D Systems, Minneapolis, MN, USA). Rabbit anti-pThr172-AMPK (sc-33524, Santa Cruz Biotechnology, Dallas, TX, USA), rabbit anti-AMPK (SAB4502329, Sigma-Aldrich, St Louis, MO, USA) and rabbit anti-NKAα2 (07-674, Millipore, Temecula, CA, USA) antibodies were used together with secondary HRP conjugated goat anti-rabbit IgG antibody (7074, Cell Signaling Technology, Danvers, MA, USA). As a loading control for nuclear protein, rabbit anti-Lamin-B1 (ab16048, Abcam, Cambridge, UK) antibody was used together with secondary Cy5 conjugated goat anti-rabbit IgG (PA45012, GE Healthcare, Little Chalfont, UK) antibody. As a loading control for total cellular protein, rabbit anti-GAPDH (2118, Cell Signaling Technology, Danvers, MA, USA) antibody was used together with secondary Cy5 conjugated goat anti-rabbit IgG antibody. HRP conjugated secondary antibodies were visualized with Western Lightning Ultra (PerkinElmer Inc, Waltham, MA, USA) chemiluminescent substrate. Blots were imaged with the Chemidoc MP imaging system (Bio-Rad, Hercules, CA, USA).

### 2.8. Tissue Collection

Four-week-old animals were sacrificed with CO_2_ inhalation followed by cervical dislocation and perfused transcardially with 6 mL phosphate buffered saline (PBS, Gibco, ThermoFisher Scientific, Waltham, MA, USA). For neonatal tissue collection, animals were sacrificed by decapitation after which the heart was removed. For RNA isolation from embryonic day 12.5 mice, female mice from timed matings were sacrificed with CO_2_ inhalation followed by cervical dislocation and the ventricular part from the extracted foetal heart was flash frozen.

### 2.9. Histology

Heart tissue was fixed in solution containing 4% paraformaldehyde and 7.5% sucrose for four hours and then incubated in 15% sucrose solution overnight. Masson’s trichrome staining was performed for 5 µm thick cross-sectional slices from four-week-old mouse ventricles and 5 µm thick longitudinal slices from neonatal mouse hearts.

### 2.10. Echocardiography

A Vevo2100 Ultrasound imaging system (VisualSonics Inc., Toronto, ON, Canada) was used for in vivo cardiac imaging as previously [6]. Mice were imaged under inhalation anaesthesia induced with 4% isoflurane (Baxter International Inc., Deerfield, IL, USA) and 400 mL/min air. Anaesthesia was maintained with 2% isoflurane and 200 mL/min air. During and shortly after imaging mice were kept on a heated platform. Cardiac parameters were assessed from short-axis M-Mode measurements.

### 2.11. Confocal Calcium Imaging

Cardiomyocytes were loaded with Fluo-4-acetoxymethyl-ester (10 μM; 0.02% pluronic acid, Invitrogen, ThermoFisher Scientific, Waltham, MA, USA) and measurements were performed with a FluoView 1000 confocal inverted microscope (Olympus, Tokyo, Japan) as previously [6]. See detailed methods in Appendix A online.

### 2.12. Patch-Clamp Recordings

Coverslips with attached cells were transferred to the recording chamber where they were perfused with DMEM or Tyrode solution (146 mM, NaCl, 4.5 mM KCl, 1.1 mM CaCl_2_, 1 mM MgCl_2_, 10 mM HEPES, 10 mM glucose, pH 7.4 adjusted with NaOH). All HEPES-buffered solutions were continuously bubbled with 100% oxygen. Carbogen gas was used for DMEM. An Axopatch 200B patch-clamp amplifier in combination with a Digidata 1440A and Clampex 10 software (Molecular Devices Inc., Sunnyvale, CA, USA) were used for the whole cell currents and action potential (AP) recordings as previously [6]. See detailed methods as well as protocols for action potential [21], sodium current [22], L-type calcium [23] current and sodium/potassium pump current [24] recordings in Appendix A online.

### 2.13. Analysis of Energy Metabolism

Isolated cardiomyocytes from 4-week-old mice were plated on Matrigel (BD Biosciences, Franklin Lakes, NJ, USA) coated XF24 cell plates (Agilent Technologies, Santa Clara, CA, USA) in XF Minimal Base Medium (Agilent Technologies, Santa Clara, CA, USA) supplemented with 2 mM GlutaMAX (Gibco, ThermoFisher Scientific, Waltham, MA, USA), 10 mM BDM and different energy substrates. The different energy substrate compositions were: 4.5 g/L glucose and 0.11 g/L sodium pyruvate (glucose + pyruvate); 4.5 g/L glucose (glucose); 0.2 mM palmitate (Sigma-Aldrich, St Louis, MO, USA) conjugated to BSA (Sigma-Aldrich, St Louis, MO, USA) (palmitate). Data were normalized to total cellular protein in wells. After addition of the supplements, the pH of the solutions was adjusted to 7.4. The oxygen consumption rate (OCR) and extracellular acidification rate (ECAR) were assessed with a Seahorse XF24 analyser (Agilent Technologies, Santa Clara, CA, USA). The maximal metabolic capacity was determined by injecting 1.5 µM carbonyl cyanide-4-(trifluoromethoxy) phenylhydrazone (FCCP) to the cells during the assay. 

To determine the effect of NKA pump inhibition on α4^+^ cardiomyocyte metabolism, isolated neonatal cardiomyocytes were plated on fibronectin-gelatine coated XF24 cell plates and kept in culture for one day. In intact cells, NKA inhibition leads to intracellular Na^+^ accumulation, which in turn activates the reverse mode of the plasmalemmal sodium-calcium exchanger leading to intracellular Ca^2+^ accumulation and cell contraction. To avoid the activation of contractile machinery during the assay, the experiment was conducted in a Ca^2+^ free solution in the presence of butanedione monoxime. One hour prior to the assay, the culture medium was changed to Tyrode solution without CaCl_2_ supplemented with 2 mM GlutaMAX, 4.5 g/L glucose, 0.11 g/L sodium pyruvate and 10 mM butanedione monoxime. Injection of 10 µM ouabain during the Seahorse assay was carried out to determine the proportion of cellular energy consumed on the more ouabain-sensitive component of the NKA pump. To quantify the proportion of NKA energy consumption from the total cellular energy consumed, data from each well was normalized to the last baseline recording of the corresponding well.

### 2.14. Adenoviral Overexpression

Data on recombinant adenovirus-encoding GFP, PGC-1α1, or PGC-1α4 have been previously published [11]. For comparing the effects of adenoviral PGC-1α1 and PGC-1α4 on adult cardiomyocyte gene expression, cells were isolated from 18-week-old mice and plated on 6-well plates. After 3-h incubation in adult cardiomyocyte plating medium, cells were exposed to virus preparations in adult cardiomyocyte culture medium. The GFP control group received GFP virus preparation, PGC-1α1 and PGC-1α4 groups received corresponding virus preparation and GFP virus preparation, and the PGC-1α1+PGC-1α4 group received PGC-1α1 and PGC-1α4 virus preparations so that each experimental group was exposed to equal total amounts of virus preparations. Isoprenaline (100 nM) was added to medium 18 h after viral transduction and RNA was isolated 6 h later (24 h after the start of transduction). For comparing the effects of adenoviral PGC-1α4 in isolated neonatal and adult cardiomyocytes, GFP and PGC-1α4 viruses were introduced to neonatal cells one day after plating, and for 18-week-old mouse cells after the 3 h incubation in plating medium. RNA was isolated 24 h after the start of transduction. Adenoviral transduction for HL-1 cardiomyocytes was performed two days after plating when cell cultures were confluent, and samples were prepared 24 h after the start of transduction.

### 2.15. RNA Library Preparation and Sequencing

Total RNA was isolated from frozen ventricular tissue samples using TissueLyzer II and TRI reagent. RNA library preparation and sequencing were performed at the Finnish Microarray and Sequencing Centre/Biocenter Finland (Turku, Finland). Fragment Analyzer (Advanced Analytical Technologies, Inc., Ankeny, IA, USA) and Bioanalyzer 2100 (Agilent, Santa Clara, CA, USA) were used to assess the RNA sample quality, and RQN/RIN values of the analysed samples were between 9.0 and 10.0. Library preparation was performed according to Illumina TruSeq^®^ Stranded mRNA Sample Preparation Guide (part # 15031047 rev E). Poly-T oligo attached magnetic beads were used to purify mRNA after which it was fragmented with divalent cations at an elevated temperature. After preparation of strand specific cDNA, adapter ligation and PCR amplification, a Bioanalyzer 2100 was used to assess the quality of the libraries and fragment size was between 200 and 700 bp with an average size of 250–350 bp. Libraries were sequenced with a HiSeq 3000 System (Illumina, San Diego, CA, USA) by pooling the 16 samples and running in two lanes. The sequencing depth of the samples was between 35 and 53 million reads. Base calling was performed with bcl2fastq software (version 2.20, Illumina, San Diego, CA, USA) with automatic adapter trimming. The acquired fastq files were used in the subsequent analyses.

### 2.16. Analysis of Chromatin Immunoprecipitation Data

Chromatin immunoprecipitation (ChIP) data for genomic occupancy of REST in differentiated C2C12 cells were acquired from the mouse ENCODE project (GEO dataset GSE36024). Reads stored in fastq files were aligned to mouse DNA (GRCm38) with bowtie2 software [25]. Peak calling was performed with HOMER software (version 4.10.3) [26] and peaks were annotated with UROPA software [27] against Ensembl’s gene annotation release version 92.

### 2.17. Culture of HL-1 Cardiomyocyte Cell Line

Experiments with HL-1 cardiomyocytes were performed at passages 65–75. Cells were cultured in HL-1 culture medium (Claycomb Medium (Sigma-Aldrich, St Louis, MO, USA) supplemented with 10% FBS, 4 mM L-glutamine, 1% PS and 100 µM norepinephrine (Sigma-Aldrich, St Louis, MO, USA)). Culture dishes were coated with 25 µg/mL fibronectin in 0.02% gelatine solution.

### 2.18. Gene Silencing with Small Interfering RNA

HL-1 cardiomyocytes were transfected with siRNA targeting Rest mRNA one day after plating when cultures were 70–80% confluent. AllStars Negative Control siRNA (Qiagen) or siRNA targeting Rest mRNA (SI01399965, Qiagen, Hilden, Germany) were transfected with RNAiMAX reagent (Invitrogen, ThermoFisher Scientific, Waltham, MA, USA) as per the manufacturer’s instructions. RNAiMAX-siRNA complex solution was diluted 1/10 in HL-1 culture medium and added to the cells. The final siRNA concentration was 30 nM.

### 2.19. Luciferase Gene Reporter Assay

Genomic sites from mouse Fam57b and Syt7 gene promoters and RE1 sites were cloned to a pGL3-basic luciferase plasmid construct (Promega, Madison, WI, USA). From the Fam57b gene promoter, a 672 bp region (from –646 bp to +26 bp of TSS), and from the Syt7 gene promoter, a 768 bp region (from –714 bp to +54 bp of TSS) were amplified by PCR and inserted upstream of luciferase using restriction enzymes MluI and XhoI. From the intronic part of the Fam57b gene, a 569 bp region (from +8487 bp to +9055 bp of TSS), and from the intronic part of Syt7 gene, a 504 bp region (from +13625 bp to +14128 bp of TSS) containing the RE1 sites were amplified by PCR and inserted downstream from luciferase into the plasmid with the corresponding gene promoter using the SalI restriction enzyme. All generated plasmids were sequenced to confirm the identity of inserted regions. PCR primers for cloning the mouse gene promoters and RE1 sites are listed in Appendix A.

HL-1 cardiomyocytes cultured on 96-well-plates were co-transfected with luciferase and pRL-tk renilla (Promega, Madison, WI, USA) plasmids one day after plating when cultures were 70–80% confluent. Lipofectamine 3000 reagent (Invitrogen, ThermoFisher Scientific, Waltham, MA, USA) was used according to the manufacturer’s instructions for transfecting cells with 0.4 µg luciferase plasmid and 0.1 µg pRL-tk plasmid constructs. The lipofectamine-plasmid complex was diluted 1/4 in HL-1 culture medium and added to the cells. For combined siRNA knockout of REST and luciferase plasmid transfection, HL-1 cardiomyocytes were first incubated with siRNA transfection reagents for 6 h, after which the solution was changed to reporter plasmid transfection reagents. For combined luciferase plasmid transfection and adenoviral PGC-1α4 overexpression, HL-1 cardiomyocytes transfected with reporter plasmids were transduced 24 h after transfection. Forty-eight hours after the reporter plasmid transfection, cells were lysed and assessed using a Dual-Luciferase^®^ Reporter Assay System (Promega, Madison, WI, USA) with the CLARIOStar microplate reader (BMG LABTECH GmbH, Ortenberg, Germany) according to the manufacturer’s instructions. Renilla luminescence in each well was used in the normalization of the luciferase signal.

### 2.20. Statistical Testing

Data are presented as means and error bars indicate SEM. In pair-wise comparisons, Student’s *t*-test was used to assess statistical significance. If three or more groups were compared, a one-way ANOVA followed by Bonferroni post hoc test were used. In case data for experimental groups were collected from several non-independent batches, a hierarchical statistical model was applied [28].

## 3. Results

### 3.1. Pgc-1α Isoforms Arising from Differential Promoter Usage and mRNA Splicing Are Expressed in Cardiac Tissue

To explore Pgc-1α isoform expression in cardiac tissue, we analyzed mouse heart RNA sequencing data [29,30,31] (Appendix A, datasets DS1–DS4). Compared to the previously published mouse *Pgc-1α1, -α2, -α3* and *-α4* (Appendix A) sequences, transcript assemblies from the mouse heart showed the presence of reads aligning to the exon 1a-exon 2 junction from the proximal promoter, and both exon 1b-exon 2 and exon 1b’-exon 2 junctions from the alternative promoter (Figure 1a and Appendix A). Furthermore, both exon 6-exon 7a and exon 6-exon 7b junctions leading to full-length and truncated isoforms, respectively, were present (Figure 1a and Appendix A). None of the mouse datasets contained reads aligning to the junction between exon 3 and exon 7, suggesting that *Pgc-1α2* and *-α3* are not expressed in the heart.

Beta adrenergic stimulation by isoprenaline in adult mouse cardiomyocytes led to a 2.7-fold increase in total *Pgc-1α* mRNA (Figure 1b). This increase was due to the activation of the alternative promoter (exon 1b/1b’) since transcripts originating from the proximal promoter (exon 1a) were unchanged (Figure 1b). Transcripts with exon 7b were induced 3.8-fold, and there was a small increase (1.4-fold) in the ratio of exon 7b-containing transcripts and total *Pgc-1α* mRNA (Figure 1b), indicating that acute β-adrenergic stimulation affected splicing between exons 6 and 7. PCR assays detecting specific exon 7b-containing gene isoforms from the cDNA of isoprenaline treated cardiomyocytes showed that these transcripts mainly arise from the alternative promoter in cardiomyocytes (Figure 1c,d). The β-adrenergic response of the alternative promoter seems to be developmentally constrained, since in early embryonic (E12.5) cardiomyocytes, where the sympathetic pathways are not fully developed, *Pgc-1α* transcripts were unaffected by isoprenaline (Appendix A). Assessed from previously published RNA-sequencing data from the adult mouse heart [32], isoprenaline administration in vivo for two weeks activated the alternative *Pgc-1α* promoter as well (Figure 1e,f). Following chronic β-adrenergic stimulation, alternative promoter activation was accompanied by decreased proximal promoter expression (Figure 1e,f), which together led to smaller total *Pgc-1α* induction (1.1-fold) compared to acute stimulation in isolated cardiomyocytes (Figure 1b). Contrary to acute stimulation (Figure 1b), there was no change in the ratio of exon 7a and 7b-containing transcripts in chronic β-adrenergic stimulation (Figure 1f).

Analyses from previously published RNA-sequencing datasets by others showed that decreased *Pgc-1α* levels are seen in both infarcted [31] (DS2) and load-induced [29] (DS3,DS4) myocardium (Figure 1g). There were no consistent changes in junctional ratios in the assessed datasets and exon 6-exon 7 junctional ratios were especially stable (Figure 1g). We also assessed human cardiac *PGC-1α* expression from RNA-sequencing data [33,34,35,36] (Appendix A, datasets DS5–DS8). Human transcript assemblies contained all the junctional reads corresponding to mouse *Pgc-1α,* except that they lacked the exon 1b’-exon 2 junction (Appendix A). Datasets allowing comparison between the healthy and diseased state did not show significant changes in total *PGC-1α* levels but, as in the mouse heart, junctional ratios did not imply large differences in promoter usage or splicing (Appendix A).

Taken together, in both mouse and human hearts the Pgc-1α gene produces isoforms containing canonical and alternative exons. Alternative splicing between exons 6 and 7 suggests that both canonical full-length and truncated PGC-1α proteins are expressed in cardiac tissue.

### 3.2. Cardiomyocyte-Specific PGC-1α4 Overexpression in Mice Leads to Early Death by Dilated Heart Failure

Since *Pgc-1α4* was expressed in cardiac tissue, and since it has previously been associated with distinct phenotypic changes in skeletal muscle cells [11] and hepatocytes [12], we decided to study its effect on cardiac physiology in a transgenic mouse model where PGC-1α4 overexpression is restricted to cardiomyocytes. The transgene was activated in ventricular tissue of PGC-1α4-overexpressing mice (α4^+^) by embryonic day 12.5 (Figure 2a), and its protein product was in the nucleus (Figure 2b). Strikingly, α4^+^ mice died a few weeks after birth (Figure 2c). Echocardiographic analysis at the age of four weeks showed no change in left ventricular (LV) wall thickness (Figure 2d, left) but increased LV inner diameter (Figure 2d, middle), indicating that overexpression induces dilated cardiomyopathy. A drastic decrease in the ejection fraction strongly suggests that the early death of α4^+^ mice is the result of heart failure (Figure 2d, right). Heart weight was increased (Figure 2e) and a cross section from the ventricles confirmed the dilated phenotype (Figure 2f). Gene expression of the beta myosin heavy chain in addition to atrial and brain natriuretic peptides were increased, and the alpha myosin heavy chain decreased, showing the typical response of α4^+^ cardiomyocytes to prevailing failure (Figure 2g). Furthermore, the endogenous PGC-1α1 protein is diminished (Figure 2b) as typical for the failing heart.

### 3.3. PGC-1α4 Overexpression Leads to Distinct Changes in Cardiomyocyte Phenotype without Effects on Energy Metabolic Capacity

To determine the cellular effects of PGC-1α4 overexpression, we next characterized the phenotype of isolated cardiomyocytes. Consistent with dilation of the ventricular walls, isolated α4^+^ cardiomyocytes had increased length and decreased width (Figure 3a,b, left and middle). Cell capacitance was increased, which indicates increased cell size (Figure 3b, right). As expected, α4^+^ cardiomyocytes showed drastic impairments in E-C coupling (Figure 3c). The amplitude and decay of electrically evoked calcium transients were decreased together with the caffeine releasable pool of the sarcoplasmic reticulum (SR) calcium. As a result of reduced SR calcium content as well as weakened calcium release and removal, α4^+^ cardiomyocytes had compromised contractility (Figure 3e). Altogether, the calcium signalling and contractile phenotype of α4^+^ cardiomyocytes are typical for end-stage heart failure.

Action potential analyses show that repolarization of α4^+^ cardiomyocytes was slower in early phases but reached the resting membrane potential (RMP) earlier (Figure 3f,g). Surprisingly, in α4^+^ cardiomyocytes, RMP was more negative, AP amplitude higher and rise time shorter than those in control cells (Figure 3h), which warranted more detailed membrane current measurements. The density of the sodium current (I_Na_) was drastically increased in α4^+^ cells (Figure 3i), which explains the shorter AP rise time compared to control cells (Figure 3h). Half-maximal voltage dependent activation of I_Na_ was reached at a more negative membrane potential in α4^+^ cells, but there were no changes in inactivation or recovery of I_Na_ (Appendix A). L-type calcium current (I_CaL_) density was increased with a leftward shift in activation and inactivation (Figure 3j and Appendix A), which is likely to contribute to the slower early AP repolarization (Figure 3g). 

Due to the well-known role of PGC-1α1 in cellular metabolism, we also measured cellular respiration from isolated α4^+^ cells. Despite other typical pathological changes, the respiratory capacity of α4^+^ cardiomyocytes was unaffected as assessed from the maximal oxygen consumption rate (OCR) of pyruvate-supplemented cells (Figure 3k and Appendix A). Basal OCR was drastically increased, which is most likely a result of pathological changes in ion homeostasis. Similar trends were also observed in extracellular acidification rate (ECAR) and OCR of cells supplemented with only glucose (Figure 3l and Appendix A) as well as OCR of cells supplemented with only palmitate (Figure 3m and Appendix A). This indicates that PGC-1α4 does not induce changes in cardiomyocyte energy substrate specificity or a tendency towards either aerobic or anaerobic energy production. AMP kinase (AMPK) activation was increased in α4^+^ ventricles as assessed from the ratio of phosphorylated and total AMPK (Appendix A), which suggests that α4^+^ cardiomyocytes also suffer from energy deprivation.

Compared to canonical PGC-1α1, isoform 4 seems to have no effect on cardiomyocyte energy-producing capacity. Observed changes in calcium transient kinetics are most likely a by-product of the severe phenotype, whereas changes in action potential properties were unexpected considering the overall pathological phenotype of the α4^+^ heart, which suggests that electrophysiological traits are specifically targeted by PGC-1α4 in cardiac myocytes.

### 3.4. PGC-1α4-Induced Gene Expression Response and the Resulting Phenotype Depend on Cardiomyocyte Developmental Stage

We set out to solve the initial gene expression changes behind the α4^+^ phenotype. The drastic cardiac pathology induced by chronic PGC-1α4 overexpression by the age of four weeks hinders solving the origins of its actions in α4^+^ cardiomyocytes. For this reason, we first measured the expression of known PGC-1α target genes in isolated adult cardiomyocytes under acute PGC-1α1 and -1α4 overexpression (Figure 4a). PGC-1α1 increased the expression of *Tfam, Cpt1b, Sod2 and Ucp3,* and except for *Tfam*, PGC-1α4 overexpression had a similar effect (Figure 4a). Combined overexpression of both isoforms blocked the PGC-1α1 induction of *Tfam* and had an incremental effect on *Ucp3* expression (Figure 4a). Next, we measured the expression of PGC-1α targets from the ventricles of 4-week-old and neonatal α4^+^ mice that we expected to have a milder phenotype compared to older animals. The only changes detected were an increase in *Cpt1b* in neonates and a substantial decrease in *Ucp3* at the age of four weeks (Figure 4b). It should be noted that the full-length PGC-1α protein was drastically reduced in 4-week-old α4^+^ ventricles (Figure 2b), which was accompanied by decreased *Pgc-1α* exon1a-containing transcripts (Figure 4b), suggesting that the repression of endogenous PGC-1α1 occurs at the transcriptional level. In neonates, where exogenous PGC-1α4 is already activated (Figure 2a), endogenous *Pgc-1α* exon1a expression is still intact (Figure 4b). This implies that the reduction in endogenous PGC-1α in 4-week-old ventricles is not a direct effect of exogenous PGC-1α4, but is likely a result of pathological mechanisms activated in the diseased α4^+^ heart.

We decided to investigate the phenotype of neonatal α4^+^ mice where pathological transcriptional remodelling has not yet manifested. During the neonatal phase (postnatal day 1–2), the observed genotype ratio did not differ from the Mendelian ratio, whereas at the time of genotyping (postnatal day 16–24) the ratio was significantly altered, with the α4^+^ genotype underrepresented (Figure 4c). This suggests that PGC-1α4 overexpression becomes detrimental within two weeks of birth. In line with the survival of neonatal α4^+^ mice, the heart morphology (Figure 4d) or calcium signalling of the isolated cardiomyocytes (Figure 4e) were not changed in comparison to the control. To get a better overall view of the effects of PGC-1α4 overexpression, we assessed gene expression from ventricular tissue of neonatal and 4-week-old mice with RNA sequencing. As expected, when compared to age-matched controls, ventricles from the 4-week-old animals had more pronounced transcriptomic changes compared to neonates (Figure 4f). Enrichment analysis shows that, in neonatal ventricles, the only consistent process targeted by PGC-1α4 is ion transport (Figure 4g and Appendix A). In the 4-week-old ventricles, in addition to ion homeostasis, several other processes related to other cardiac functions and extracellular structures are enriched, which most likely results from the pathological changes induced at this time point (Figure 4g and Appendix A).

RNA sequencing analysis did not indicate that chronic PGC-1α4 overexpression in α4^+^ ventricles induces drastic or consistent changes in energy metabolism, which contrasts with the acute adenoviral overexpression in cultured adult cells where PGC-1α4 seems to induce traditional PGC-1α1 metabolic targets (Figure 4a). This led us to compare the gene expression changes between neonatal and adult cardiomyocytes during acute adenovirus-mediated PGC-1α4 overexpression. We chose 25 genes related to cardiomyocyte function and energy metabolic pathways that were upregulated in neonatal α4^+^ ventricles, of which 15 were also induced in 4-week-old ventricles (Figure 4h). In acute adenoviral overexpression, 19 of these genes were upregulated in neonatal cells whereas in adult cardiomyocytes only four genes were induced (Figure 4i). These data indicate that in cardiomyocytes, the PGC-1α4 effect depends on the developmental stage. Moreover, it seems that in the α4^+^ mice, the transcriptional changes induced in developing cardiomyocytes continue into postnatal stages and do not reflect the effects of acute PGC-1α4 induction in adult cardiomyocytes.

### 3.5. PGC-1α4 Overexpression Induces Na-K ATPase Expression and Leads to a Drastic Increase of Its Current in Neonatal Cardiomyocytes

Because RNA sequencing showed enrichment of genes encoding for ion transport proteins in α4^+^ ventricles already at the neonatal stage (Figure 4g), we decided to measure action potentials from neonatal ventricular cardiomyocytes, as it might reveal the primary effects of PGC-1α4 overexpression (Figure 5a). As opposed to the cells from 4-week-old animals, in neonatal cardiomyocytes repolarization (Figure 5b), amplitude and rise time of the action potential (Figure 5c, middle and right) were unchanged, but the resting membrane potential (Figure 5c, left) was more negative and similar to 4-week-old α4^+^ cells (Figure 3h, left). We inspected the expression of individual transcripts encoding for ion transporters to identify changes from which the electrophysiological phenotype could originate. Indeed, the expression of several different sarcolemmal ion transporters changed in α4^+^ ventricles already at neonatal phases (Appendix A). One of the most prominent changes was the induction of sodium-potassium ATPase (NKA) isoforms α2 and α3 (*Atp1a2, Atp1a3*) whose transcripts reach the level of the major neonatal isoform NKAα1 (*Atp1a1*, Appendix A). Since NKA is electrogenic with a net outward current during diastole at physiological potassium and sodium concentrations [37], the changes in its current density might well explain the changes seen in RMP (Figure 5c, left).

Neonatal α4^+^ cardiomyocytes showed increased expression of NKA isoforms α2 and α3 and total NKA current (Figure 5d–f). Different NKA isoforms in rodent cardiomyocytes have specific sensitivity to its inhibitor ouabain [38,39]. As expected, in α4^+^ cardiomyocytes where NKAα2 and NKAα3 proteins are induced, the proportion of more ouabain-sensitive NKA current component is increased (Figure 5g). Maintenance of sodium and potassium gradients across the cell membrane by the NKA pump is one of the most energy-consuming processes in cardiac myocytes. In line with NKA current measurements, the rate of glycolysis was reduced in neonatal α4^+^ cardiomyocytes upon the addition 10 µM ouabain, whereas in control cells, glycolysis remained unchanged (Figure 5h), suggesting that increased NKA current leads to higher energy consumption in neonatal α4^+^ cardiomyocytes.

Functional data from neonatal cardiomyocytes together with gene pathway analysis suggests that the α4^+^ heart phenotype originates from the changes in membrane ion homeostasis during early development, which leads to pathological changes within the two-week period after birth.

### 3.6. Genes Associated with the Transcriptional Repressor REST Are Induced by PGC-1α4 in Developing Cardiac Myocytes

We employed the RNA sequencing data from α4^+^ hearts in search of possible cellular pathways that PGC-1α4 targets in cardiomyocytes. In addition to ion homeostasis, enriched GO biological processes and cellular compartments included those related to neuronal function (Appendix A). Since many of the transcripts in these neuronal gene groups were upregulated in α4^+^ hearts, we decided to investigate if PGC-1α4 somehow interferes with RE1-Silencing Transcription factor (REST, aka Neuron-Restrictive Silencer Factor (NRSF)), a repressor of neuronal gene expression in cardiomyocytes [40]. We used expression profiles of a previously reported mouse model with heart-specific REST knockout to determine a collection of genes repressed by REST in the heart [41]. We narrowed down the list of 192 upregulated genes in the REST knockout heart to 134 by only including genes that were associated with REST DNA binding in the C2C12 cell line (Appendix A). Our α4^+^ RNA sequencing dataset contained 45 of the REST target genes, of which 19 were upregulated in α4^+^ ventricles (Figure 6a,b). Almost all of these 19 genes had the highest expression levels in neuronal tissues in mouse (Appendix A), which further implies that they are neuronal genes normally repressed by REST in cardiomyocytes. Gene expression of *Rest* and other members of the REST complex was downregulated at the age of four weeks but remained unchanged in neonatal α4^+^ ventricles (Appendix A), indicating that the enrichment of REST target genes in the neonatal heart is not caused by reduced expression of REST complex members. The expression of REST targets induced in neonatal α4^+^ ventricles was already increased at embryonic day 12.5 (Appendix A), showing that PGC-1α4 is already active during early cardiac development. Like other genes induced in α4^+^ ventricles (Figure 4h,i), genes identified as REST targets showed induction in neonatal but not in adult cardiomyocytes under acute PGC-1α4 overexpression (Figure 6c). In adult cardiomyocytes, seven out of the 14 selected REST target genes were undetected in both control and overexpressing cells (Figure 6c), suggesting that the effect of PGC-1α4 via REST is restricted to genes typically expressed in early cardiomyocytes.

Out of the seven REST targets induced in neonatal α4^+^ ventricles, adenoviral PGC-1α4 overexpression induced the expression of three (*Atp1a3, Fam57b* and *Syt7*) in HL-1 cardiomyocytes (Figure 6d). The *Cdkn1a* gene that has been reported to be repressed by REST in the heart [41] was also induced. siRNA-mediated knock-down of REST led to increased expression of *Atp1a3, Syt7* and *Cdkn1a*, confirming their repression by REST in HL-1 cells (Figure 6e). Next, we constructed reporter plasmids that have the genomic sites containing a RE1 consensus sequence under the ChIP-sequencing peaks associated with *Syt7* and *Fam57b* (Appendix A) downstream and the corresponding gene promoters upstream of the luciferase gene. Both *Syt7* and *Fam57b* promoter-containing plasmids induced robust luciferase activity in HL-1 cells, which was decreased by the downstream RE1 site (Appendix A). REST knockdown rescued the luciferase activity in RE1-site containing plasmids (Appendix A) confirming the interaction of REST with the cloned RE1 sites. PGC-1α4 overexpression partially rescued *luciferase* expression that was dampened by the RE1 site (Figure 6f), confirming the interference with REST repression by PGC-1α4.

### 3.7. Actions of Different PGC-1α Isoforms during Normal Cardiomyocyte Development Are Governed by the Expression of Their Target Transcription Factors

As the effects of PGC-1α4 on REST-regulated genes are restricted to developing cardiomyocytes (Figure 6c), we decided to investigate their expression during normal development. Ventricular expression of *Atp1a3, Atp2b2* and *Bdh1* is robustly decreased in neonates when compared to embryonic day 12.5 (Figure 7a). *Atp1a3* and *Atp2b2* remain low, but *Bdh1* is reactivated by the age of four weeks (Figure 7a). On the other hand, expression of *Fam57b* and *Syt7* were increased in later time points compared to E12.5 (Figure 7a). To determine if the expression of these genes is under the control of PGC-1α4/REST in early cardiomyocytes, we measured their mRNA levels from the ventricles of cardiomyocyte-specific PGC-1α knockout mice at E12.5. Downregulation of *Atp1a3, Atp2b2* and *Bdh1* in knockout ventricles (Figure 7b) suggests that PGC-1α4-mediated REST de-repression is responsible for their activation in early cardiomyocytes. As has been previously reported [42], cardiac expression of RESTdecreased very soon after birth (Figure 7c), indicating that the PGC-1α4/REST interaction can only occur during early development. Concurrent with decreasedREST, *Ppara* and *Pparg* were increased and a tendency towards the splicing of full-length *Pgc-1α* isoforms was increased as the ratio of exon 7b containing transcripts and total *Pgc-1α* mRNA decreased in neonates and 4-week-old mice compared to E12.5 (Figure 7c). Similar expression patterns for PGC-1α binding partners in the mouse heart were assessed from a previously published microarray dataset comprising time points from early development into adulthood (Figure 7d). In addition to *Ppara* and *Pparg*, *oestrogen-related receptor alpha (Esrra)* was increased, whereas *Esrrg* and *nuclear respiratory factor 1 (Nrf1)* expression levels remained relatively stable throughout development (Figure 7d).

Collectively, the results of our study suggest that during normal cardiomyocyte development, the effects of the main PGC-1α isoforms on their target gene expression are dependent on the presence of their binding partners. As the window for PGC-1α4/REST interaction closes in the perinatal phase, the expression of traditional PGC-1α coactivated transcription factors such as PPARs is increased, enabling the well-known metabolism-promoting effects of PGC-1α (Figure 7e).

## 4. Discussion

Heart contraction is one of the most energy-demanding processes in the body, and therefore, cardiac function and cardiomyocyte energy metabolism are inherently coupled. In early embryonic cardiomyocytes, where the maturation of excitation–contraction coupling has just been initiated [43], a considerable amount of ATP is required for the still very immature ion homeostasis and contraction [44]. In the functionally and metabolically mature adult heart, the capacity to produce ATP can further increase in adaptation to exercise [45]. On the other hand, pathologies of the heart are almost invariably associated with depressed energy metabolism [46]. For these reasons, different ways to improve cardiomyocyte metabolic capacity have been investigated in the search for novel cardiac therapies. Since its discovery, peroxisome proliferator-activated receptor (PPAR) gamma coactivator 1-alpha (PGC-1α) has been actively studied in the cardiac context as its activity in cardiomyocytes is strongly associated with their energy producing capacity [2]. In addition to the canonical PGC-1α1, other protein isoforms have been described [10] but their expression and function has not been studied extensively in cardiac myocytes.

In the present study, we show that in the heart, due to alternative promoter activation and differential exon splicing, mRNAs encoding for different protein isoforms are produced from the PGC-1α gene. PGC-1α4, a truncated isoform encoded by a transcript arising from the alternative promoter, regulates RE1-Silencing Transcription factor (REST) activity and induces the expression of neuronal genes in developing cardiomyocytes. REST normally represses genes associated with neuron function in non-neuronal cells via binding to its DNA response element at gene regulatory regions [47]. REST is essential for cell type specification in early development and mice, as either germ-line [48] or cardiac-specific [41] REST knockout mice display an embryonic lethal phenotype associated with ectopic neuronal gene expression. REST is also an epigenetic modifier acting as a so-called pioneer transcription factor that opens condensed chromatin to the poised state, allowing the binding of other transcription factors to activate gene expression [49,50]. In cardiac cells there are two distinct modes how REST regulated genes behave during development; they can either be permanently silenced during the prenatal phase or become induced postnatally as REST expression decreases after birth [42]. Interestingly, our analysis indicated that PGC-1α4 induces REST targets whose expression is both decreased (*Atp1a3, Atp2b2*) and increased (*Fam57b, Syt7*) during normal development. This implies that PGC-1α4 specifically targets the REST complex rather than some other transcriptional modifiers responsible for the differential temporal changes in REST target expression.

In developing cardiomyocytes, the PGC-1α4 mediated induction of *Atp1a3* and *Atp2b2*, encoding for sarcolemmal Na^+^/K^+^ and Ca^2+^ pumps, respectively, is expected to increase cellular energy consumption as both proteins are ATPases that can move ions against their electrochemical gradient across the cell membrane [37,51]. Indeed, NKA pump capacity was robustly increased in neonatal α4^+^ cardiomyocytes, which also led to an increase in their energy consumption. This finding suggests that when *Pgc-1α* transcription is activated during cardiomyocyte development, PGC-1α1 enhances ATP production and concurrently the alternative isoform PGC-1α4 induces changes in ATP consumption. A similar effect is seen in skeletal muscle where PGC-1α4 induces myocyte hypertrophy and improves endurance capacity [11]. Since heart tissue is particularly sensitive to an imbalance between energy production and consumption, it is not a surprise that disturbing the normal expression pattern of the two PGC-1α isoforms has a deleterious effect. Similarly, excessive expression of the PGC-1α isoform 1 has been shown to be detrimental for cardiac function [5], whereas moderate isoform 1 overexpression induces exercise-like changes in cardiomyocytes [6]. It is noteworthy that *Atp1a3* and *Atp2b2* expression is normally restricted to prenatal cardiomyocyte development, but in the α4^+^ heart, they are erroneously expressed after birth. This is likely one of the reasons why the α4^+^ phenotype becomes detrimental, because postnatal cardiac growth requires a significant amount of cellular energy [52]. If a large amount of ATP is consumed by the misregulated plasmalemmal ATPases, it might hinder normal growth, leading to the observed dilated phenotype. The energetic state of the α4^+^ heart might be similar to patients with dilated cardiomyopathy where reduced metabolic capacity, seen in reduced myocardial phosphocreatine/ATP Ratio, has been shown to predict a decrease in the survival rate of the patients [53].

Acute PGC-1α4 overexpression in isolated wild-type adult cardiomyocytes induces the same targets as PGC-1α1, which is in line with early studies on N-terminal isoforms showing that they can activate the same transcription factors as the canonical isoform [54,55]. Since the alternative promoter is induced by β-adrenergic stimulation in adult cardiomyocytes, induction of PGC-1α transcripts originating from the distal site efficiently couples energy production and increased contractile activity in situations where the β-adrenergic pathway is activated. This could be especially important in the heart where activity of the proximal promoter seems to be particularly stable. Chronic in vivo and acute in vitro PGC-1α4 overexpression results are conflicting in that only chronic overexpression can induce REST targets in adult cardiomyocytes. It should be noted that there is low REST expression remaining in postnatal hearts, and since REST is an epigenetic modifier, it could be that constant PGC-1α4 overexpression potentiates the remnant REST protein so that its DNA binding capacity remains during the perinatal phase in α4^+^ cardiomyocytes. Persistent expression of REST-regulated genes in α4^+^ hearts postnatally further implies that in certain pathological situations, the REST protein could also be activated in the adult heart and PGC-1α4 might participate in the induction of neuronal genes that are associated with heart pathologies [56].

Our results update the role of PGC-1α in heart development. In cardiomyocyte physiology, the canonical isoform, PGC-1α1, is an established component in the maturation and maintenance of postnatal energy metabolism. In the developing heart, PGC-1α4 seems to have no effect on cardiomyocyte energy-producing capacity but it has an unexpected functional impact on E-C coupling via REST. Since REST expression decreases during cardiomyocyte development [42], it is not a surprise that the transcription-inducing effect of PGC-1α4 on REST-repressed genes disappears perinatally. In contrast, the traditional PGC-1α1 target transcription factors, such as PPARs and ESRRα, are induced after birth. This suggests that during normal cardiomyocyte development the expression level of PGC-1α isoforms is not a determinant of cellular function per se, but rather it is the presence of their binding partners that governs their functional impact.

To conclude, we expand the role of PGC-1α in cardiac physiology by identifying a novel molecular pathway in developing cardiomyocytes regulated by the non-canonical isoform PGC-1α4. Functional changes induced by PGC-1α4 via REST increase energy consumption, which suggests that supply and demand of cellular energy in cardiomyocytes are coupled by different PGC-1α isoforms.

## Figures and Tables

**Figure 1 cells-11-02944-f001:**
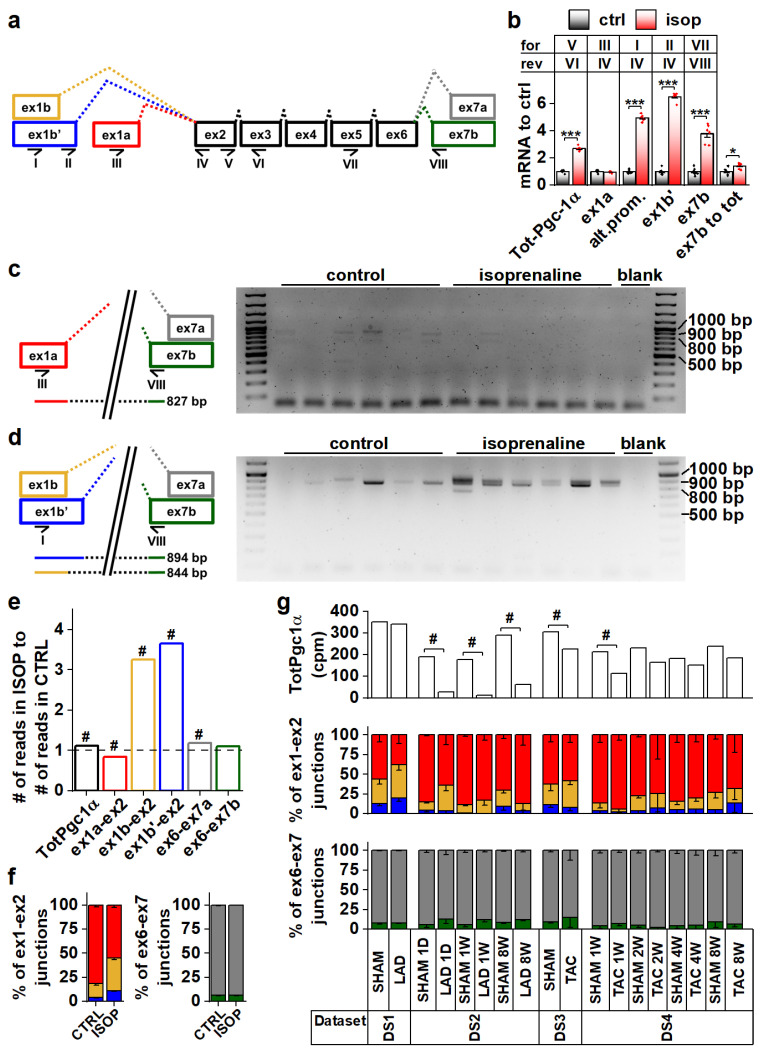
*Pgc-1α* isoforms arising from differential promoter usage and mRNA splicing are expressed in cardiac tissue. (**a**) Schematics showing the different *Pgc-1α* exons and splicing events present in the mouse heart. Primers detecting expression of specific exons are identified by Roman numerals. Colour coding of exon-exon junctions is used in subsequent panels presenting RNA sequencing data. (**b**) RT-qPCR analysis of total and exon expression of *Pgc-1α* in adult cardiomyocytes exposed to 100 nM isoprenaline for four hours (*n* = 6). Names of the primers (for/rev) used in each assay refer to panel a. (**c**) Gel electrophoresis from PCR reaction detecting expression of N-terminal Pgc-1α from the proximal promoter (*NT-Pgc-1α-a*) from isoprenaline-exposed adult mouse cardiomyocyte cDNA. Schematic on left shows how the PCR product is produced from the transcript. (**d**) Gel electrophoresis from PCR reaction detecting expression of N-terminal Pgc-1α from the alternative promoter (*Pgc-1α4* and *NT-Pgc-1α-c*) from isoprenaline-exposed adult mouse cardiomyocyte cDNA. Schematic on left shows how PCR products are produced from the transcripts. (**e**) Ratio of total *Pgc-1α* expression and individual exon-exon junctions in hearts of adult mice treated with isoprenaline compared to vehicle control (*n* = 56). Bar colours of exon-exon junction ratios refer to colours in panel a. (**f**) Ratios of RNA sequencing reads aligning to different exon 1-exon 2 (left) and exon 6-exon 7 (right) junctions in hearts of adult mice administered with vehicle control and isoprenaline (*n* = 56). Colours of bars refer to colours in panel a. (**g**) Total *Pgc-1α* expression (counts per million total reads, cpm) (top), ratios of reads aligning to exon 1-exon 2 junctions (middle) and ratios of reads aligning to exon 6-exon 7 junctions (bottom) assessed from publicly available RNA sequencing datasets from different mouse cardiac disease models. Colours of bars in middle and top refer to colours in panel a. Student’s *t*-test *p*-value: *, *p* < 0.05; ***, *p* < 0.001. DESeq2 analysis *p*-value: # < 0.01.

**Figure 2 cells-11-02944-f002:**
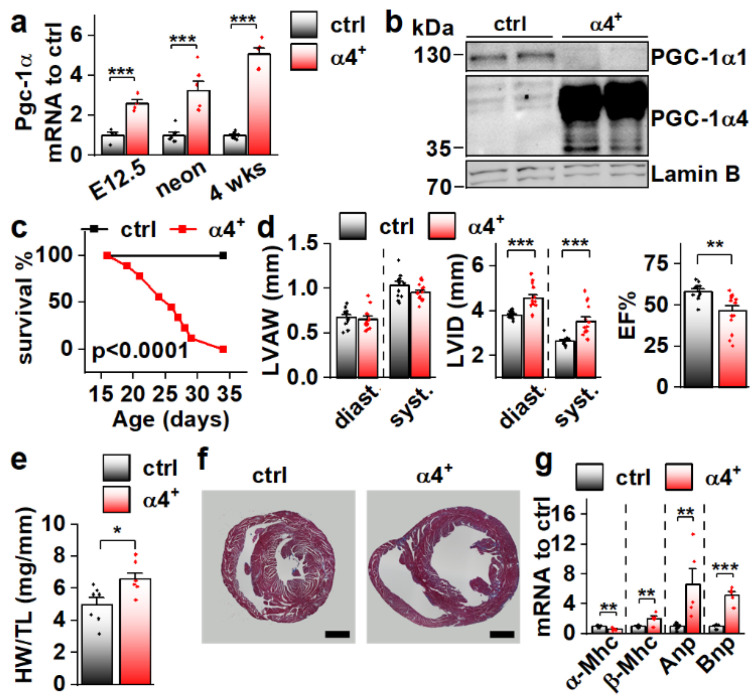
Cardiomyocyte-specific PGC-1α4 Overexpression in Mice Leads to Early Death by Dilated Heart Failure. (**a**) Total Pgc-1α gene expression in the ventricles of embryonic day 12.5 (E12.5, *n*[ctrl] = 4, *n*[α4^+^] = 4), neonatal (neon, *n*[ctrl] = 6, *n*[α4^+^] = 6) and 4-week-old (*n*[ctrl] = 7, *n*[α4^+^] = 5) Pgc-1α4 overexpressing mice (α4^+^). (**b**) Protein expression of PGC-1α isoforms 1 and 4 in the nuclear fraction from left ventricular tissue of control and α4^+^ mice at the age of four weeks. (**c**) Survival of control (*n* = 13) and α4^+^ (*n* = 9) mice after genotyping. Statistical significance was determined using a Log-rank (Mantel–Cox) test. (**d**) Left ventricular anterior wall thickness (LVAW, left), left ventricular inner diameter (LVID, middle) and Ejection fraction (EF, right) assessed with echocardiography in 4-week-old animals (*n*[ctrl] = 12, *n*[α4^+^] = 14). (**e**) Heart weight to tibia length (HW/TL) (*n*[ctrl] = 7, *n*[α4^+^] = 7). (**f**) MTC staining of transversal cardiac cross-sections at the age of four weeks (scale 1 mm). (**g**) Gene expression of heart failure markers in left ventricular tissue at the age of four weeks (*n*[ctrl] = 7, *n*[α4^+^] = 5). Student’s *t*-test *p*-value: *, *p* < 0.05; **, *p* < 0.01; ***, *p* < 0.001.

**Figure 3 cells-11-02944-f003:**
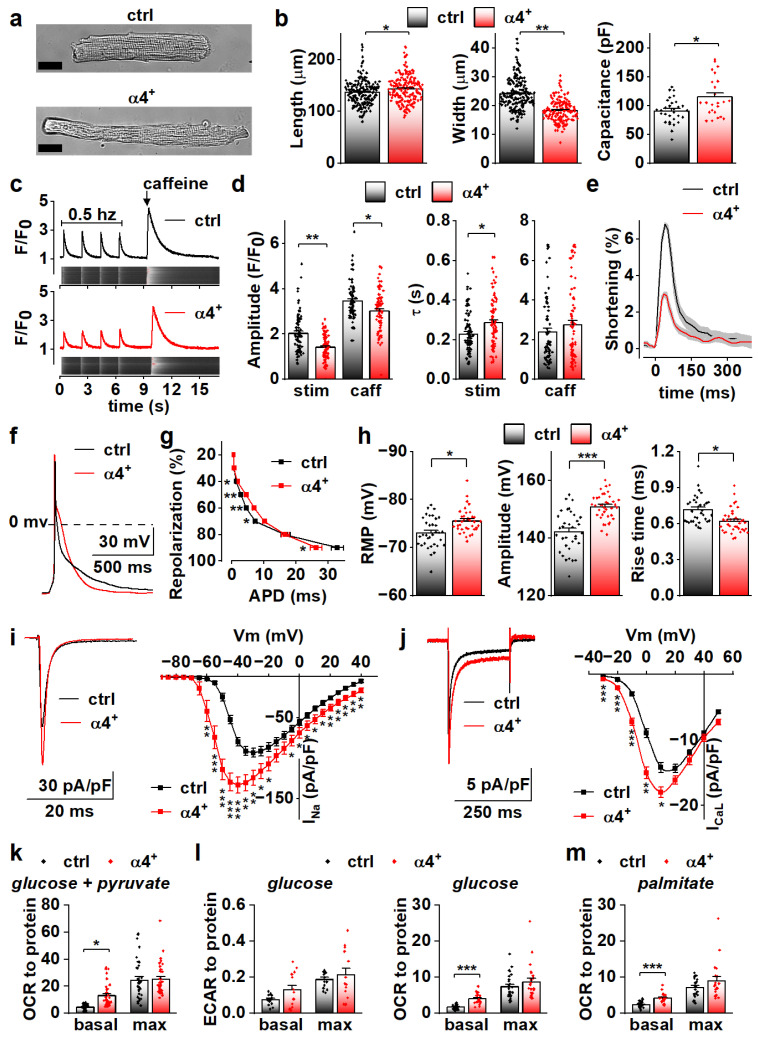
PGC-1α4 Overexpression Leads to Distinct Changes in Cardiomyocyte Phenotype. (**a**) Representative light microscopic images from isolated control (ctrl) and Pgc-1α4-overexpressing (α4^+^) cardiomyocytes. Scale bar 20 µm. (**b**) Length, width (left, middle, *n*[ctrl] = 2/178 (animals/cells), *n*[α4^+^] = 2/166) and membrane capacitance (right, *n*[ctrl] = 5/30, *n*[α4^+^] = 3/24) of isolated cardiomyocytes. (**c**) Representative calcium transients from Fluo-4 loaded cardiomyocytes. (**d**) Amplitudes (left) and decay times (right) of the stimulation-evoked (stim) and caffeine pulse (caff) calcium transients (*n*[ctrl] = 4/77, *n*[α4^+^] = 5/79). (**e**) Averaged shortening of imaged cardiomyocytes caused by stimulation (*n*[ctrl] = 37, *n*[α4^+^] = 9, grey area represents SEM). (**f**) Representative action potentials from isolated control and α4^+^ cardiomyocytes. (**g**) Action potential duration (APD) plotted against repolarization phase (*n*[ctrl] = 4/31, *n*[α4^+^] = 4/35). (**h**) Resting membrane potential (RMP, left), amplitude (middle) and rise time (right) of the action potential (*n*[ctrl] = 4/34, *n*[α4^+^] = 4/38). (**i**) Representative traces (left) and current–voltage curves (right, *n*[ctrl] = 5/26, *n*[α4^+^] = 3/23) of sodium current. (**j**) Representative traces (left) and current–voltage curves (right, *n*[ctrl] = 3/19, *n*[α4^+^] = 3/14) of L-type calcium current. (**k**) Basal and maximal (max, induced with FCCP) oxygen consumption rates (OCR) in isolated cardiomyocytes supplemented with glucose and pyruvate (*n*[ctrl] = 5/33 (animals/seahorse wells), *n*[α4^+^] = 5/38). (**l**) Basal and maximal extracellular acidification rates (ECAR, left, *n*[ctrl] = 5/17, *n*[α4^+^] = 6/15) and OCR (*right, n*[ctrl] = 5/25, *n*[α4^+^] = 6/24) in isolated cardiomyocytes supplemented with glucose. (**m**) Basal and maximal OCR in isolated cardiomyocytes supplemented with palmitate (*n*[ctrl] = 5/23, *n*[α4^+^] = 6/20). *p*-value from hierarchical statistical test: *, *p* < 0.05; **, *p* < 0.01; ***, *p* < 0.001.

**Figure 4 cells-11-02944-f004:**
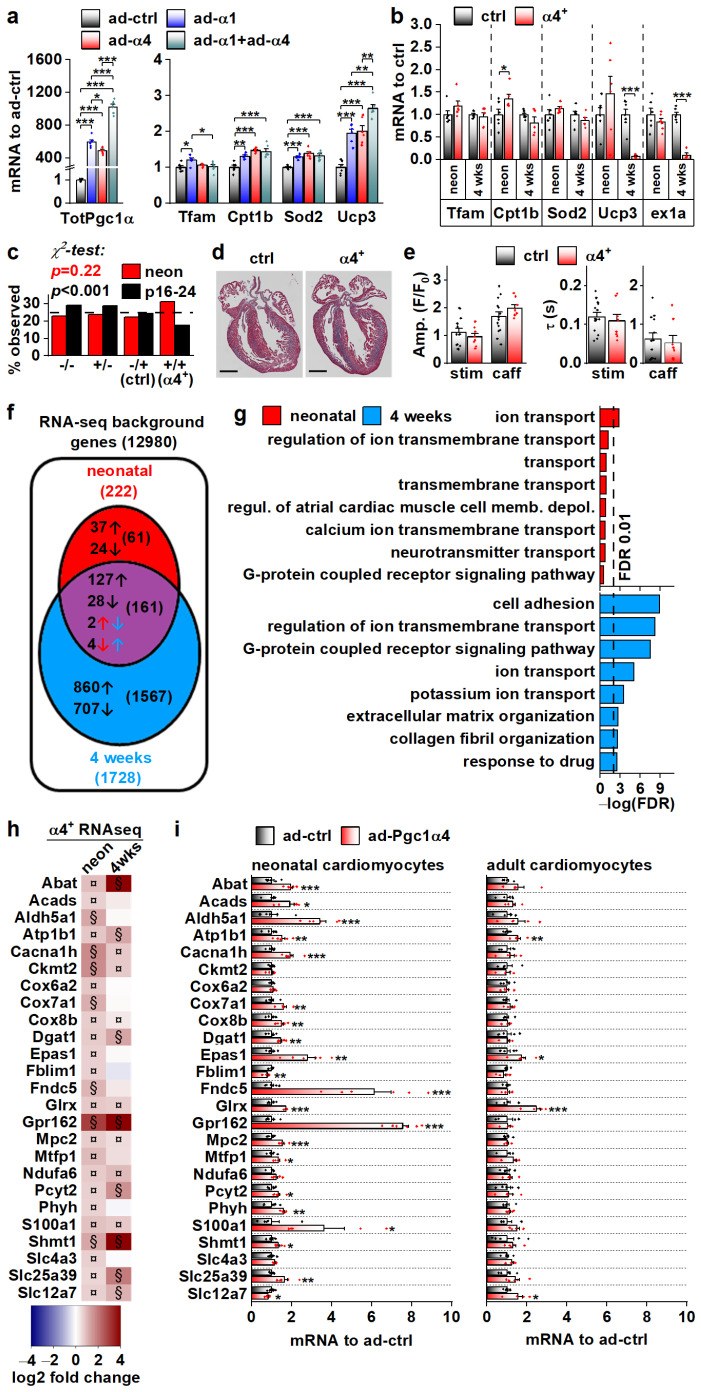
PGC-1α4-induced Gene Expression Response and the Resulting Phenotype Depend on Cardiomyocyte Developmental Stage. (**a**) Expression of total *Pgc-1α* and known PGC-1α1 target genes in cultured adult cardiomyocytes transduced with adenoviruses expressing Pgc-1α1 and Pgc-1α4 (*n* = 6). (**b**) Expression of known PGC-1α1 target genes and exon 1a (ex1a) containing *Pgc-1α* transcripts in the ventricles of neonatal (neon, *n*[ctrl] = 6, *n*[α4^+^] = 6) and 4-week-old (*n*[ctrl] = 7, *n*[α4^+^] = 5) Pgc-1α4-overexpressing mice (α4^+^). (**c**) Ratios of live mice with different combinations Myh6-Cre/Flox-Pgc-1α4 transgenes 1–2 days (neon) and 16–24 days after birth (*n*[neon] = 215, *n*[p16–24] = 577). (**d**) MTC staining of longitudinal cardiac cross-sections from one-day-old control (ctrl) and α4^+^ heart (scale 1 mm). (**e**) Amplitudes (left) and decay times (right) of the stimulation-evoked (stim) and caffeine pulse (caff) calcium transients in cultured neonatal cardiomyocytes (*n*[ctrl] = 2/14, *n*[α4^+^] = 2/8). Blue and orange lines represent mean and distribution of the data. (**f**) Gene expression changes in ventricles of neonatal and 4-week-old Pgc-1α4 overexpressing (α4^+^) mice (*n* = 4) as assessed with RNA sequencing. Genes with absolute logarithmic fold change above 1 and adjusted *p*-value below 0.01 are considered differentially expressed. (**g**) Top eight enriched biological processes (gene ontology) among the differentially expressed genes in neonatal and 4-week-old α4^+^ ventricles. (**h**) Heatmap showing log_2_ fold change in comparison to age-matched control in selection of genes induced in neonatal α4^+^ ventricles in neonatal and 4-week-old α4^+^ ventricles (*n* = 4). Note that genes with absolute log_2_ fold change below 1 were not included in enrichment analyses in panels f and g. (**i**) Gene expression in selection of genes induced in neonatal α4^+^ ventricles in cultured neonatal and adult cardiomyocytes transduced with PGC-1α4 adenovirus (*n*[adeno neon] = 5–6, *n*[adeno adult] = 4–5). Student’s *t*-test *p*-value (or Bonferroni post hoc test *p*-value for one-way ANOVA in panel a or hierarchical test *p*-value in panel e): *, *p* < 0.05; **, *p* < 0.01; ***, *p* < 0.001. Heatmap: §, DESeq2 analysis *p*-value *p* < 0.01 and absolute log2 fold change > 1; ¤, DESeq2 analysis *p*-value *p* < 0.01 and absolute log2 fold change < 1.

**Figure 5 cells-11-02944-f005:**
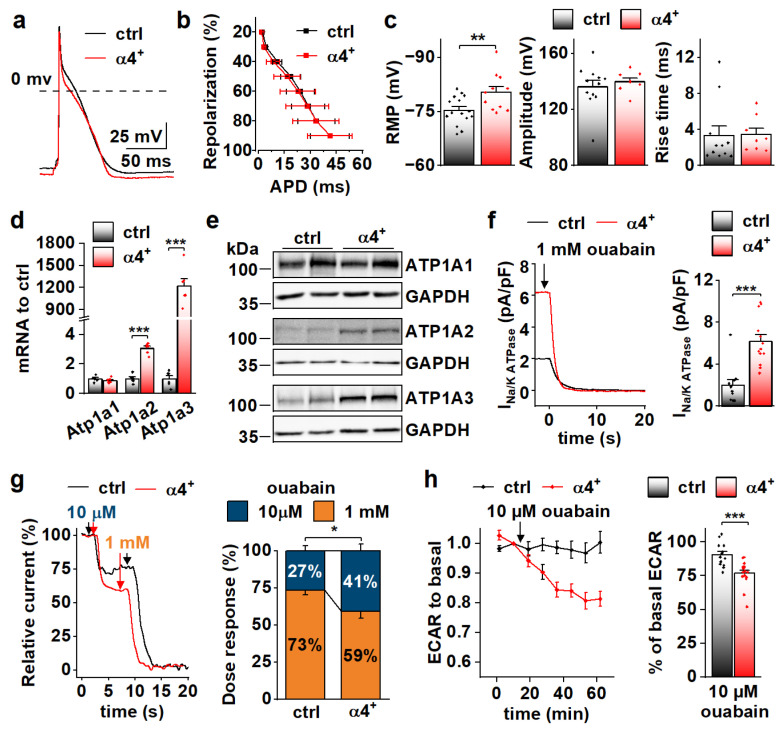
PGC-1α4 Overexpression Induces Na-K ATPase Expression and Leads to a Drastic Increase in Its Current in Neonatal Cardiomyocytes. (**a**) Representative action potentials from cultured neonatal cardiomyocytes. (**b**) Action potential duration (APD) plotted against repolarization phase. (**c**) Resting membrane potential (RMP, left, *n*[ctrl] = 4/15 (animals/cells), *n*[α4^+^] = 3/11), amplitude (middle, *n*[ctrl] = 4/12, *n*[α4^+^] = 3/8) and rise time (right, *n*[ctrl] = 4/11, *n*[α4^+^] = 3/8) of the action potentials. (**d**) mRNA expression of sodium–potassium ATPase isoforms α1 (Atp1a1), α2 (Atp1a2) and α3 (Atp1a3) in ventricles of neonatal control (ctrl) and Pgc-1α4 overexpressing (α4^+^) (*n* = 6). (**e**) Protein expression of sodium–potassium ATPase isoforms in neonatal ventricles. (**f**) Representative Na+/K+ ATPase current traces under application of a saturating concentration of ouabain (left) and statistics of the ouabain sensitive current (right) in isolated neonatal cardiomyocytes (*n*[ctrl] = 3/11, *n*[α4^+^] = 2/13). (**g**) Representative relative Na^+^/K^+^ ATPase current traces under consecutive applications of 10 µM and 1 mM ouabain (left) and statistics of relative proportion of the current sensitive to 10 µM ouabain (right) in neonatal cardiomyocytes (*n*[ctrl] = 1/7, *n*[α4^+^] = 1/10). (**h**) Changes in glycolytic rate of isolated neonatal cardiomyocytes induced application of 10 µM ouabain (*n*[ctrl] = 3/14 (animals/wells), *n*[α4^+^] = 3/17). *p*-value from hierarchical statistical test (or Student’s *t*-test in panel d): *, *p* < 0.05; **, *p* < 0.01; ***, *p* < 0.001.

**Figure 6 cells-11-02944-f006:**
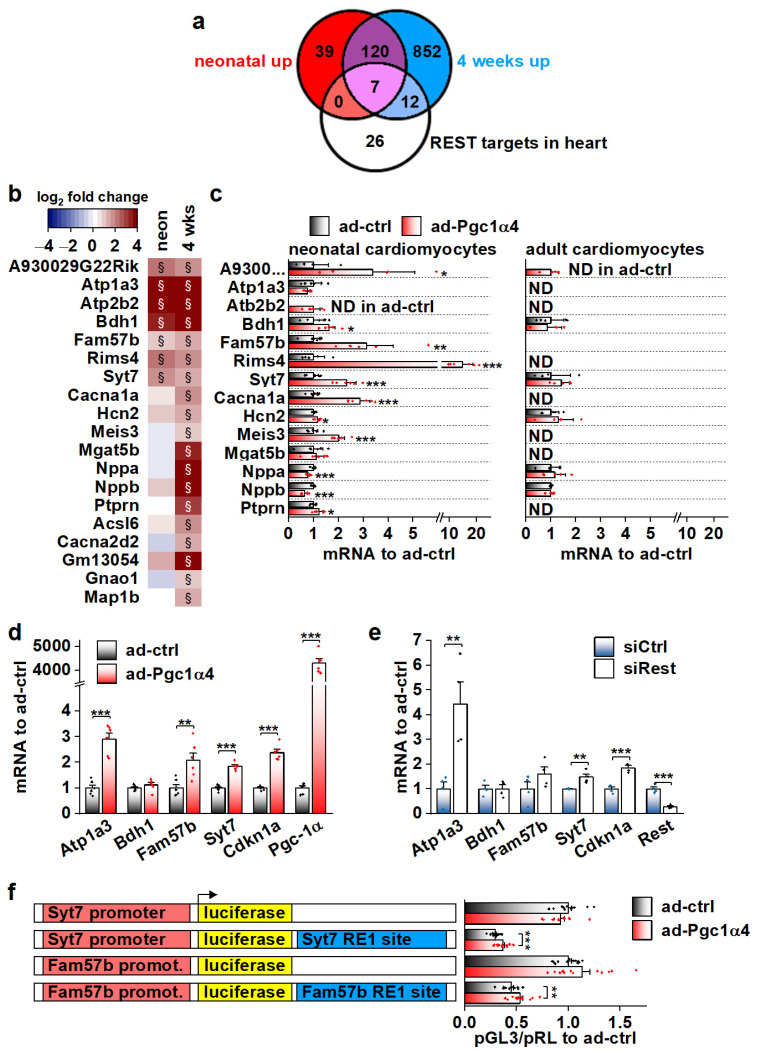
Genes associated with transcriptional repressor REST are induced by PGC-1α4 in cardiac myocytes. (**a**) Venn diagram showing the overlap of REST target genes with genes upregulated in α4^+^ ventricles. (**b**) Heatmap showing log2 fold change of REST target genes in comparison to control in neonatal and 4-week-old α4^+^ ventricles (*n* = 4). Please note that genes with absolute log_2_ fold change below 1 were not included in enrichment analyses in panel a. (**c**) Gene expression of REST target genes in cultured neonatal and adult cardiomyocytes transduced with PGC-1α4 adenovirus (*n*[adeno neon] = 5–6, *n*[adeno adult] = 4–5). (**d**) Gene expression of selected REST target genes in adenoviral PGC-1α4 transduced HL-1 cells (*n* = 6). (**e**) Gene expression of selected REST target genes in REST siRNA-treated HL-1 cells (*n* = 4). (**f**) Luciferase activity in adenoviral PGC-1α4 transduced HL-1 cells transfected with pGL3-basic plasmids containing endogenous Syt7 and Fam57b promoters upstream and with corresponding RE1 sites downstream of *luciferase* (*n* = 12 from 3 independent experiments). Student’s *t*-test *p*-value (or hierarchical statistical test *p*-value in panel f): *, *p* < 0.05; **, *p* < 0.01; ***, *p* < 0.001. Heatmap: §, DESeq2 analysis *p*-value *p* < 0.01 and absolute log_2_ fold change >1.

**Figure 7 cells-11-02944-f007:**
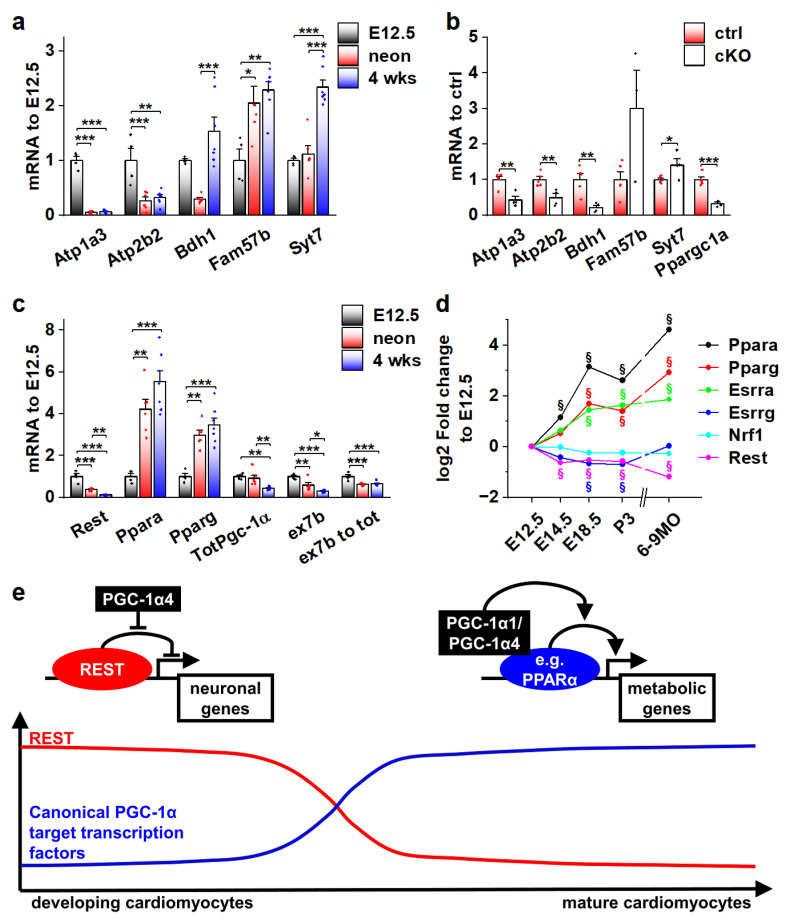
Actions of different PGC-1α isoforms during normal cardiomyocyte development are controlled by the expression of their binding partners. (**a**) Gene expression of selected REST target genes in α4^+^ control mouse ventricles at different ages (*n*[E12.5] = 4, *n*[neonatal] = 6, *n*[four weeks] = 7). (**b**) Gene expression of selected REST target genes in PGC-1α knockout mouse ventricles at E12.5 (*n*[ctrl] = 4, *n*[ko] = 5). (**c**) Gene expression of PGC-1α and its binding partners in α4^+^ control mouse ventricles at different ages (*n*[E12.5] = 4, *n*[neonatal] = 6, *n*[four weeks] = 7). (**d**) Gene expression of PGC-1α binding partners in α4^+^ control mouse ventricles during embryonic and postnatal development. (**e**) Schematics of PGC-1α isoform actions on different target transcription factors during cardiomyocyte development. Student’s *t*-test *p*-value (or Bonferroni post hoc test *p*-value for one-way ANOVA in panels **a** and **b**): *, *p* < 0.05; **, *p* < 0.01; ***, *p* < 0.001. §, adjusted *p*-value < 0.01 in microarray analysis.

## Data Availability

RNA sequencing data produced in this study is stored in NCBI’s GEO database under accession number GSE213642. Previously published data sets analysed in this study were downloaded from NCBI’s GEO database (BioProject identifiers: PRJNA227375, PRJNA472253, PRJNA470729, PRJNA277489, PRJNA445706, PRJNA291619, PRJNA280990, PRJEB8360 and PRJNA318773).

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
