# Peer review of "PGC-1α4 Interacts with REST to Upregulate Neuronal Genes and Augment Energy Consumption in Developing Cardiomyocytes"

_cells, 2022, doi:10.3390/cells11192944_

Round 1

Reviewer 1 Report

The paper entitled “PGC-1α4 interacts with REST to upregulate neuronal genes and 2 augment energy consumption in developing cardiomyocytes” describes the role of PGC1-a4 in energy metabolism of cardiomyocytes in a developmentally distinct, stage-specific manner. The authors used a wide range of techniques to unravel the role of non-canonical PGC1-a in cardiomyocytes physiology. The experiments are well designed and performed. The results are explained thoroughly. But there are some questions which I would like to be addressed by the authors.

My major point deals with the type of transgene. Why overexpression of PGC1-a. If the focus of the study was to address the function of this protein isoform, the first choice should have been knockout mouse. I would like to ask the authors to explain why mice with overexpression of PGC1-a4 was used?

Other points are listed as below:

1.     The materials and methods is written very well and with great details. This is really appreciated. But I suggest to keep main methods in the manuscript and transfer details into supplementary. For instance, keep patch clamp method here and transfer single current recordings to supplementary.

2.     Why in patch clamp recording, 1.1 mM Ca in external solution. It should be at least 1.8 mM.

3.     In Na-K ATPase current recording, internal Ca should be 1.5 * 10-8 M.

4.     Please introduce the abbreviations in the first place. For instance, chromatin immunoprecipitation in the materials and methods that should be introduced as ChIP abbreviation.

5.     In Results, figure 1g, what does cpm stand for?

6.     Why PGC1-a1 is absent in transgenic mouse (figure 2B)?

7.      In Results, section 3-4, what does acute overexpression mean? did you want to say acute stimulation by isoprenaline?

8.     Regarding PGC1-a4 and REST relation and their developmentally distinct, stage-specific interaction, the luciferase assay might not have been adequate to determine the actual partners. Why the authors did not use the co-immunoprecipitation?

9.     In Discussion, authors stated that “During prenatal development, PGC-1α4 seems to have no effect on cardiomyocyte energy-producing capacity but it has an unexpected functional impact on E-C coupling via REST”. How they could make this conclusion from their results?  

Author Response

The paper entitled “PGC-1α4 interacts with REST to upregulate neuronal genes and 2 augment energy consumption in developing cardiomyocytes” describes the role of PGC1-a4 in energy metabolism of cardiomyocytes in a developmentally distinct, stage-specific manner. The authors used a wide range of techniques to unravel the role of non-canonical PGC1-a in cardiomyocytes physiology. The experiments are well designed and performed. The results are explained thoroughly. But there are some questions which I would like to be addressed by the authors.

My major point deals with the type of transgene. Why overexpression of PGC1-a. If the focus of the study was to address the function of this protein isoform, the first choice should have been knockout mouse. I would like to ask the authors to explain why mice with overexpression of PGC1-a4 was used?

Other points are listed as below:

  1. The materials and methods is written very well and with great details. This is really appreciated. But I suggest to keep main methods in the manuscript and transfer details into supplementary. For instance, keep patch clamp method here and transfer single current recordings to supplementary.

We have moved detailed descriptions of the following methods to supplementary materials:

- 2.4. Cardiomyocyte isolation and culture

- 2.11. Confocal calcium imaging

- 2.12. Patch-clamp recordings

  1. Why in patch clamp recording, 1.1 mM Ca in external solution. It should be at least 1.8 mM.

Our patch clamping methods are based on previously published protocols by others. Buffer solution compositions are not necessarily same as in physiological fluids but, rather, they are optimized to give the best quality patch clamp recordings. Lower calcium concentration in the Tyrode solution for perfusion is used to prevent calcium overloading of the isolated cells prior to recordings.

  1. In Na-K ATPase current recording, internal Ca should be 1.5 * 10-8 M.

This is now fixed (method in the supplementary materials).

  1. Please introduce the abbreviations in the first place. For instance, chromatin immunoprecipitation in the materials and methods that should be introduced as ChIP abbreviation.

We have now checked the use of abbreviations in the manuscript.

  1. In Results, figure 1g, what does cpm stand for?

cpm stands for counts per million of sequencing reads. Abbreviation is now explained in the figure legend.

  1. Why PGC1-a1 is absent in transgenic mouse (figure 2B)?

The decrease of the endogenous PGC-1α1 is most likely resulting from the severe heart failure seen in the transgenics. To explain this we added following sentence to the end of the section 3.2.:

“Also, the endogenous PGC-1α1 protein is diminished (Figure 2b) as typical for the failing heart.”

  1. In Results, section 3-4, what does acute overexpression mean? did you want to say acute stimulation by isoprenaline?

In that specific paragraph we are comparing the gene expression of transgenic mouse (chronic PGC-1α4 overexpression) to adenovirally transduced cultured cardiomyocytes one day after transduction (acute PGC-1α4 overexpression). To clarify, we have added “acute adenoviral” to all occasions when mentioning acute overexpression.

  1. Regarding PGC1-a4 and REST relation and their developmentally distinct, stage-specific interaction, the luciferase assay might not have been adequate to determine the actual partners. Why the authors did not use the co-immunoprecipitation?

Positive result from Co-IP showing PGC-1α4 binding REST or some other member of the REST complex would indeed make the connection between PGC-1α4 and REST stronger than the luciferase assay. Unfortunately, currently available PGC-1α antibodies are not very reliable, which means that there is a high risk of getting false negative results from PGC-1α Co-IP.

Secondly as PGC-1α is a transcriptional co-activator the actual REST affecting mechanism might be mediated from an unexpected interaction. For this reason we think determining the binding partner in this case would require using tagged protein together with mass spectrometry protein identification and we do not have access to this kind of application at the moment.

For these reasons, we did not perform the Co-IP in the present study and leave the proof of the interaction at the level of changes in REST target gene expression and the luciferase assay.

  1. In Discussion, authors stated that “During prenatal development, PGC-1α4 seems to have no effect on cardiomyocyte energy-producing capacity but it has an unexpected functional impact on E-C coupling via REST”. How they could make this conclusion from their results?

Start of this sentence was poorly phrased. We have changed “During prenatal development…” to “In developing heart…” to better describe our findings.

Reviewer 2 Report

The manuscript by Tuomainen et al. investigates the differential expression of PGC-1α isoforms in the heart at different stages of development and explore the consequences of PGC-1α4 overexpression in the heart. This manuscript comprises a tour the force investigation on the role of PGC-1α4 and draws well substantiated conclusions on isoform abundance, effects on REST regulated genes during development and the impacts on energy consumption. These are strong suits to this manuscript; nevertheless, the authors found that overexpression of  PGC-1α4 induces a dilated cardiomyopathy phenotype with heart failure and the mechanistic by which PGC-1α4 overexpression causes a dilated cardiomyopathy with heart failure are not very well explored. I would greatly benefit the manuscript a profound discussion on the causative role of PGC-1α4 overexpression in dilated cardiomyopathy. Production of new data to substantiate the mechanistic role of PGC-1α4 in DCM would require a substantial amount of additional work and would expand this manuscript beyond the proposed scope, therefore the authors might consider to follow up on this mechanism in a subsequent manuscript, but they must provide a strong discussion tying cell energetics and electrophysiology to the structural changes observed in their transgenic model. 

Keep up the good work!

Author Response

The manuscript by Tuomainen et al. investigates the differential expression of PGC-1α isoforms in the heart at different stages of development and explore the consequences of PGC-1α4 overexpression in the heart. This manuscript comprises a tour the force investigation on the role of PGC-1α4 and draws well substantiated conclusions on isoform abundance, effects on REST regulated genes during development and the impacts on energy consumption. These are strong suits to this manuscript; nevertheless, the authors found that overexpression of  PGC-1α4 induces a dilated cardiomyopathy phenotype with heart failure and the mechanistic by which PGC-1α4 overexpression causes a dilated cardiomyopathy with heart failure are not very well explored. I would greatly benefit the manuscript a profound discussion on the causative role of PGC-1α4 overexpression in dilated cardiomyopathy. Production of new data to substantiate the mechanistic role of PGC-1α4 in DCM would require a substantial amount of additional work and would expand this manuscript beyond the proposed scope, therefore the authors might consider to follow up on this mechanism in a subsequent manuscript, but they must provide a strong discussion tying cell energetics and electrophysiology to the structural changes observed in their transgenic model.

Keep up the good work!

We would like to thank the reviewer for the comments and notes on our manuscript.

We agree that it would be interesting to study the mechanism leading from PGC-1α4 overexpression to dilated phenotype, but as stated by the reviewer, it would be beyond the scope of the present study where we aimed to found out the cellular effects of PGC-1α4 in cardiomyocytes. It is noteworthy that in our PGC-1α4 overexpressing mouse model the PGC-1α4 expression in cardiomyocytes rises to very high level early in the development. We believe that this PGC-1α4 misregulation, that leads to dilated phenotype soon after birth, is an extremely non-physiological situation, which is why we do not want to emphasize too much the end-point pathology of the transgenic model in the discussion. Rather, we want to highlight the findings revealing novel features of PGC-1α4 in developing cardiomyocytes. It is true that PGC-1α4 overexpressing mouse could be used to study molecular pathways related to pathological cardiac energy deprivation in general, but we think further work aiming to answer these questions would benefit from a model that is better mimicking a real pathological situation.

To tie cell energetics to structural phenotype, we have modified the discussion on page 20 in lines 734-743 as follows:

It is noteworthy, that Atp1a3 and Atp2b2 expression is normally restricted to prenatal car-diomyocyte development but in the α4+ heart they are erroneously expressed after birth. This is likely one of the reasons why the α4+ phenotype becomes detrimental because postnatal cardiac growth requires a significant amount of cellular energy[48]. If a large amount of ATP is consumed by the misregulated plasmalemmal ATPases it might hinder normal growth leading to the observed dilated phenotype. The energetic state of the α4+ heart might be similar to patients with dilated cardiomyopathy where reduced metabolic capacity, seen in reduced myocardial phosphocreatine/ATP Ratio, has been shown to predict a decrease in the survival of the patients[49].

Reviewer 3 Report

"PGC-1alpha-4 interacts with REST to upregulate neuronal genes and augment energy consumption in developing cardiomyocytes" by Tuomainen et al. explores the role of a splice form of PGC-a in cardiac development.  The authors use many different approaches to address this issue, including in vivo and in vitro studies and functional studies using overexpression and loss of function studies. Electrophysiological studies are very thorough and show ample data points.

This work is complete.  I honestly could not think of another experiment I would have wanted them to perform.   Some of the data is a little confusing, but I feel that the authors could address this with revisions and further explanations in the text rather than additional experiments.

My comments for the authors are as follows:

In figure 1, it is confusing that the total mRNA for PGC1a is less than the mRNA from the alternative promoter.  I believe that the authors are not looking at mRNA per se but rather the fold change in mRNA between the control and isoprenaline-treated samples. It would be clearer if this were stated clearly in the text.

Is figure 1b quantification of figures 1 c and d or is figure 1 b based on crossing point data?  

It is unclear from the text if figure 1g is based on previously published studies by this group or if it is publicly available data.  If it is public data, the authors should indicate how they know which splice variants are represented in each of the data sets analyzed, presumably other groups would not be using the same primer sets as the authors used.

In figure 2a, the data seems to be normalized to the control, yet there are also error bars and individual data for the controls. This isn't very clear.  If the control is set to a value of 1, why is there a range of values for the control ?  Please explain.

Figure 3. Data is nicely presented, and sufficient sample measurements are given for each data point.

Throughout the remainder of the text. Again there is this issue with the RT-PCR data being both, normalized to the control, and also showing a range of mRNA values for the control group.

In general the explanation for how the real time PCR data is quantified (in the Materials and Methods section is insufficient.

Author Response

"PGC-1alpha-4 interacts with REST to upregulate neuronal genes and augment energy consumption in developing cardiomyocytes" by Tuomainen et al. explores the role of a splice form of PGC-a in cardiac development.  The authors use many different approaches to address this issue, including in vivo and in vitro studies and functional studies using overexpression and loss of function studies. Electrophysiological studies are very thorough and show ample data points.

This work is complete.  I honestly could not think of another experiment I would have wanted them to perform.   Some of the data is a little confusing, but I feel that the authors could address this with revisions and further explanations in the text rather than additional experiments.

My comments for the authors are as follows:

Authors would like to thank the reviewer for pointing out the flaws in the gene expression data presentation.

1)

In figure 1, it is confusing that the total mRNA for PGC1a is less than the mRNA from the alternative promoter.  I believe that the authors are not looking at mRNA per se but rather the fold change in mRNA between the control and isoprenaline-treated samples. It would be clearer if this were stated clearly in the text.

Yes, we are reporting all RT-qPCR results from each individual primer set as fold change to control group (we state in the y-axis title “mRNA to ctrl”). The absolute values from different sets of primers are not necessarily comparable since individual primer pair properties can affect the signal produced from its transcript target (this means that the transcript quantities achieved with qRT-PCR are somewhat arbitrary). For this reason, we use the generally accepted practice to normalize the results to control group. We have modified the description of qRT-PCR in the methods to clarify our data presentation.

Regarding the result in Figure 1b, since isoprenaline is activating only the alternative promoter it makes sense that the alternative promoter fold change is greater than that of total mRNA because the change in the total mRNA is diluted by the presence transcripts from the unchanged proximal promoter (ex1a).

2)

Is figure 1b quantification of figures 1 c and d or is figure 1 b based on crossing point data? 

Figure 1b shows gene expression results collected with RT-qPCR using the primers depicted in Figure 1a. In Figure 1b the exon 7b (ex7b) expression signal could possibly include exon 7b containing transcripts originating from both proximal (exon 1a) and alternative promoter. To validate from which promoter the exon 7b containing transcripts are originating, we performed gel electrophoresis of products of conventional PCR using primers landing on exon 7b and either on proximal or alternative promoter. Products of PCR detecting specific exon 7b containing transcripts are too long (over 800 bp) for analysis with RT-qPCR method. We have clarified the text describing the method and the results.

3)

It is unclear from the text if figure 1g is based on previously published studies by this group or if it is publicly available data.  If it is public data, the authors should indicate how they know which splice variants are represented in each of the data sets analyzed, presumably other groups would not be using the same primer sets as the authors used.

Figure 1g (as well as Figures 1e-f) is based on the analysis of previously published RNA sequencing data by others. RNAseq data allows the analysis of individual exon expression independent of where the data was collected. It is true that it is not possible to directly compare our isoprenaline RT-qPCR results (Figure 1b) to isoprenaline RNAseq data by others (Figures 1e-f), but they both show activation of alternative promoter, which is the main result we want to point out here. We have added the references of the individual RNAseq data to the results text to highlight that the data is published by others.

4)

In figure 2a, the data seems to be normalized to the control, yet there are also error bars and individual data for the controls. This isn't very clear.  If the control is set to a value of 1, why is there a range of values for the control ?  Please explain.

We perform the normalization to control group in qRT-PCR data so that we calculate the mean of the control group data and then divide all individual data points (including control group samples) with the control group mean. This way control group mean becomes 1, and its relative variation stays the same as in the non-normalized raw data, and we can perform the statistical testing reliably. We have modified the description of qRT-PCR in the methods to clarify our data normalization.

5)

Figure 3. Data is nicely presented, and sufficient sample measurements are given for each data point.

6)

Throughout the remainder of the text. Again there is this issue with the RT-PCR data being both, normalized to the control, and also showing a range of mRNA values for the control group.

In general the explanation for how the real time PCR data is quantified (in the Materials and Methods section is insufficient.

Based on the reviewer comments 1) - 4) we have modified the manuscript text describing PCR methods and parts of the description of results.

Round 2

Reviewer 1 Report

I have no further comments for this round.

Reviewer 2 Report

I am satisfied with the authors response and their change in the discussion of the manuscript. Keep up with the good work.

Reviewer 3 Report

The authors have addressed all of my concerns about the text